# Contextual and cross-modality modulation of auditory cortical processing through pulvinar mediated suppression

Xiao-lin Chou[1,2†], Qi Fang[1,2†], Linqing Yan[1†], Wen Zhong[1], Bo Peng[1,2], Haifu Li[1], Jinxing Wei[1], Huizhong W Tao[1,3]*, Li I Zhang[1,3]*

[1]Zilkha Neurogenetic Institute, University of Southern California, Los Angeles, United States; [2]Graduate Program in Neuroscience, University of Southern California, Los Angeles, United States; [3]Department of Physiology and Neuroscience, Keck School of Medicine, University of Southern California, Los Angeles, United States

**Abstract** Lateral posterior nucleus (LP) of thalamus, the rodent homologue of primate pulvinar, projects extensively to sensory cortices. However, its functional role in sensory cortical processing remains largely unclear. Here, bidirectional activity modulations of LP or its projection to the primary auditory cortex (A1) in awake mice reveal that LP improves auditory processing in A1 supragranular-layer neurons by sharpening their receptive fields and frequency tuning, as well as increasing the signal-to-noise ratio (SNR). This is achieved through a subtractive-suppression mechanism, mediated largely by LP-to-A1 axons preferentially innervating specific inhibitory neurons in layer 1 and superficial layers. LP is strongly activated by specific sensory signals relayed from the superior colliculus (SC), contributing to the maintenance and enhancement of A1 processing in the presence of auditory background noise and threatening visual looming stimuli respectively. Thus, a multisensory bottom-up SC-pulvinar-A1 pathway plays a role in contextual and cross-modality modulation of auditory cortical processing.

**\*For correspondence:**
htao@usc.edu (HWT);
liizhang@usc.edu (LIZ)

[†]These authors contributed equally to this work

**Competing interests:** The authors declare that no competing interests exist.

## Introduction

Thalamus is generally considered as a gate to the cerebral cortex. First-order thalamic nuclei, such as the dorsal lateral geniculate nucleus (dLGN) and ventral medial geniculate body (MGBv), relay bottom-up sensory information to primary sensory cortices, visual and auditory, respectively (*Kremkow and Alonso, 2018*; *Winer et al., 2005*). They serve as the major driver of sensory responses in the cortex for each respective sensory modality (*Guillery and Sherman, 2002*; *Halassa and Sherman, 2019*). Compared to first-order thalamic nuclei, the functional roles of higher-order thalamic nuclei in sensory processing are much less well understood, although many of them are known to have broad connections with both primary and secondary cortices.

The lateral posterior nucleus (LP) of the thalamus is the rodent homologue of the primate pulvinar nucleus (*Harting et al., 1972*; *Harting et al., 1973*). The latter is considered the largest thalamic complex (*Harting et al., 1973*). Previous studies on LP/pulvinar have mostly been focused on its involvement in visual-related functions, such as visual attention, visually guided behaviors and eye movements (*Dominguez-Vargas et al., 2017*; *Saalmann et al., 2012*; *Soares et al., 2017*; *Stitt et al., 2018*; *Zhou et al., 2016*), largely due to its extensive reciprocal connections with visual cortical areas (*Beltramo and Scanziani, 2019*; *Bennett et al., 2019*; *Juavinett et al., 2020*; *Kaas and Lyon, 2007*; *Nakamura et al., 2015*; *Oh et al., 2014*; *Roth et al., 2016*; *Shipp, 2001*; *Stitt et al., 2018*; *Wong et al., 2009*; *Zhou et al., 2018*). Besides visual cortices, LP/pulvinar also has connections with other sensory cortices including primary and secondary auditory cortices

(*Cappe et al., 2009*; *Hackett et al., 1998*; *Nakamura et al., 2015*; *Oh et al., 2014*; *Shipp, 2007*) and contains auditory responsive neurons which exhibit short-latency responses (*Chalupa and Fish, 1978*; *Gattass et al., 1978*; *Magariños-Ascone et al., 1988*; *Woody et al., 1991*; *Yirmiya and Hocherman, 1987*). However, the influence of LP on auditory cortical responses has rarely been examined. The impact of LP activity on the primary auditory cortex (A1) is of particular interest. It is known that the axonal projections from LP to primary sensory cortices mainly terminate in layer 1 (L1), whereas those to secondary sensory cortices mainly terminate in L4 (*Fang et al., 2020*; *Roth et al., 2016*; *Zhou et al., 2018*). As L1 contains predominantly inhibitory neurons (*Jiang et al., 2013*; *Mesik et al., 2019*; *Schuman et al., 2019*), it is likely that the LP input to the primary sensory cortex plays some modulatory roles.

In the present study, we investigated the effect of LP activity on auditory responses of neurons in superficial layers of A1 using bidirectional optogenetic manipulation approaches. We found that LP activity improved auditory processing functions in A1 in that it helped to sharpen frequency tuning of A1 L2/3 pyramidal neurons and to enhance the signal-to-noise ratio (SNR) of their auditory responses, through subtractive suppression of their responses in analogous to a thresholding effect. In addition, we found that such effect could play a role in modulating A1 responses under noise background. Furthermore, we found that LP, by receiving salient input from SC, could also mediate the cross-modality modulation of A1 responses by visual looming stimuli.

## Results

### Bidirectional modulation of frequency tuning and SNR in A1 L2/3 by LP activity

To study whether LP could influence auditory cortical processing, we applied optogenetic approaches to manipulate LP activity and examined changes in functional response properties of A1 neurons using single-cell loose-patch recordings in awake mice (see Materials and methods). For reversible and temporal silencing of neuronal spikes, we injected adeno-associated virus (AAV) encoding ArchT into LP and implanted an optic fiber over LP to deliver green LED light (*Figure 1A*). Recordings were made in the ipsilateral A1. To investigate auditory information processing functions, we presented a set of tone pips of varying frequencies and intensities (50 ms duration, 2–45 kHz, 10–70 dB sound pressure level or SPL) and recorded tone-evoked spike responses from L2/3 pyramidal neurons (see Materials and methods). A typical L2/3 pyramidal neuron exhibited a V-shaped tonal receptive field (TRF) with a distinct characteristic frequency (CF) (*Figure 1B*, left). We interleaved LED-on and LED-off trials so that TRFs without and with LP silencing could be compared in the same neuron. When LP was silenced, we observed an overall increase in response level and TRF size as compared to the LED-off condition (*Figure 1B*, right; *Figure 1—figure supplement 1A–B*). Nevertheless, the overall shape of frequency tuning (*Figure 1C*) and the CF of TRF (*Figure 1—figure supplement 2A*) were largely preserved. Analyzing all the recorded L2/3 neurons, we found significant increases in the spontaneous firing rate (FR), evoked FR and bandwidth of frequency tuning (as measured at an intensity level of 60 dB SPL and at 20 dB above the intensity threshold) when silencing LP (*Figure 1D–F*; *Figure 1—figure supplement 3A*). More importantly, the signal-to-noise ratio (SNR), as measured by the ratio of evoked to spontaneous FR, was reduced (*Figure 1G*). These results indicated that when LP activity was suppressed auditory information processing in A1 might be compromised due to broadening of frequency tuning and a reduction in SNR.

Next, we optogenetically activated LP by injecting AAV encoding channelrhodopsin2 (ChR2) into LP and delivering blue LED light pulses (*Figure 1H*). As shown by an example L2/3 pyramidal neuron, activation of LP decreased the amplitude of tone-evoked responses and shrank the TRF (*Figure 1I–J*; *Figure 1—figure supplement 1C–D*), without changing the CF (*Figure 1—figure supplement 2B*). Opposite to the effects of silencing LP, activation of LP reduced the spontaneous and evoked FR, and narrowed the frequency tuning bandwidth, while enhancing the SNR (*Figure 1K–N*; *Figure 1—figure supplement 3B*). Therefore, the frequency tuning and SNR of A1 L2/3 pyramidal neurons could be bidirectionally modulated by manipulating LP activity: increasing LP activity enhances frequency selectivity and SNR of auditory responses and thus generally improves auditory cortical processing, and vice versa for decreasing LP activity. As a control, we performed similar experiments in GFP-expressing animals and did not observe any significant changes in auditory

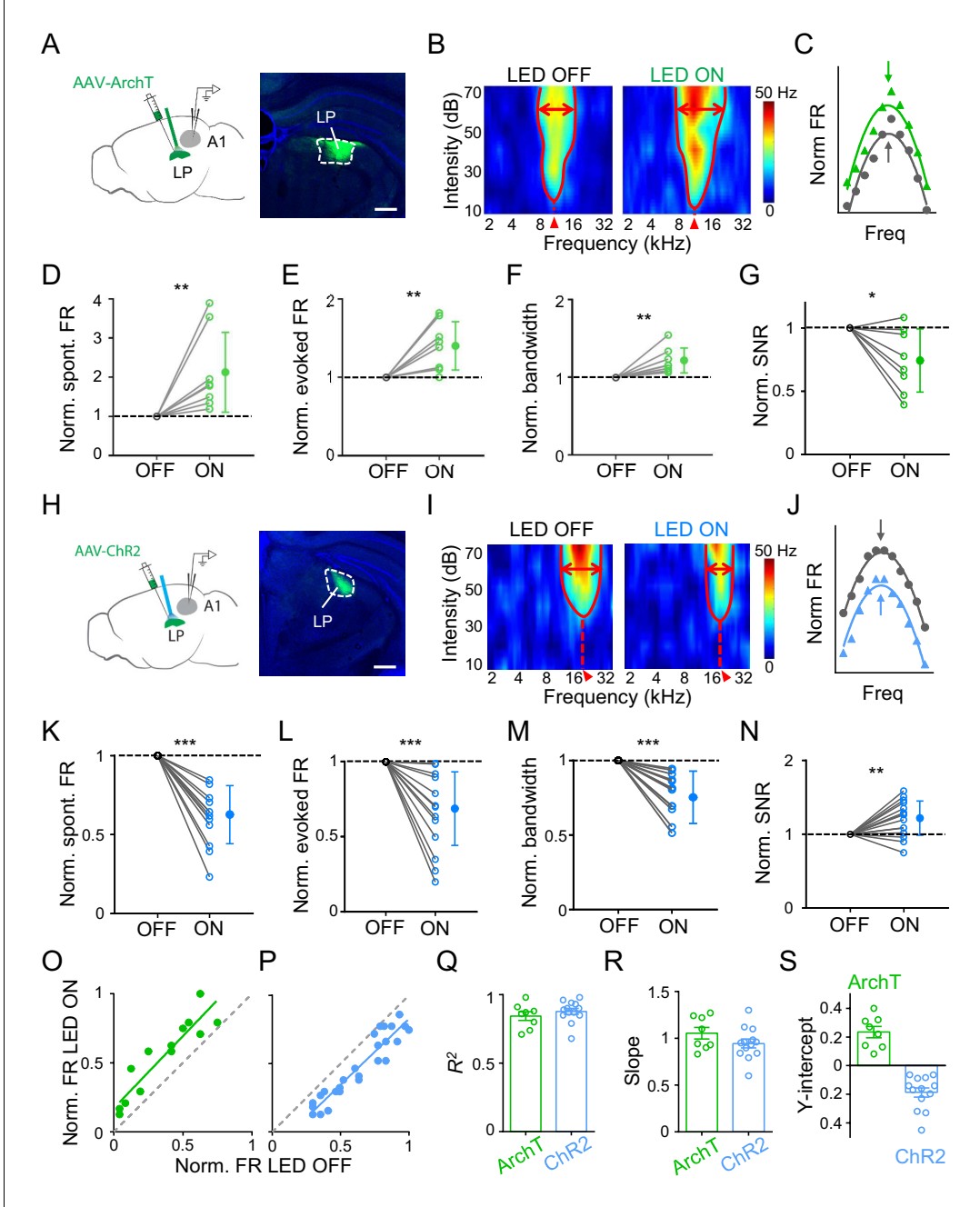

**Figure 1.** Effects of bidirectional manipulation of LP activity on A1 response properties. (**A**) Left, schematic experimental condition. AAV-ArchT was injected into LP and green LED was applied via an implanted optic fiber to silence LP neurons. Loose-patch recordings were made from A1 neurons on the same side. Right, confocal image showing the expression of ArchT within LP (marked by white dashed curve). Scale bar, 500 µm. (**B**) Reconstructed tonal receptive field (TRF) of an example A1 L2/3 pyramidal neuron without (left) and with (right) optogenetic silencing of LP. Color scale indicates evoked firing rate. Red curve marks the TRF boundary. Red double arrows depict the frequency tuning bandwidth at 60 dB SPL. Red arrowhead marks the characteristic frequency (CF). (**C**) Normalized frequency tuning curves (at 60 dB SPL) without (black) and with (green) LP silencing. Arrows mark the best frequency. (**D–G**) Normalized spontaneous firing rate (**D**, p=0.0078, Wilcoxon signed-rank test), evoked firing rate (**E**, p=0.0079, paired *t*-test), tuning bandwidth at 60 dB SPL (**F**, p=0.0069, paired *t*-test) and signal-to-noise ratio (**G**, p=0.022, paired *t*-test) of recorded A1 L2/3 neurons without (OFF) and with (ON) LP silencing. **p<0.01, *p<0.05, n = 8 cells in 4 animals. Data points for the same cell are connected with a line. (**H**) Left, experimental condition. AAV-ChR2 was injected into LP and blue LED was delivered to activate LP neurons. Right, example image showing the expression of ChR2 within LP. Scale bar, 500 µm. (**I**) TRF of an example A1 L2/3 pyramidal neuron without (left) and with (right) optogenetic activation of LP neurons. (**J**) Normalized frequency tuning curves without (black) and with (blue) LP activation. (**K–N**) Normalized spontaneous firing rate (**K**, ***p<0.001, paired *t*-test), evoked firing rate (**L**, p<0.001, paired *t*-test), tuning bandwidth at 60 dB SPL (**M**, p<0.001, paired *t*-test), and SNR (**N**,

*Figure 1 continued on next page*

*Figure 1 continued*

p=0.0059, paired *t*-test) of A1 L2/3 neurons without and with LP activation (n = 13 cells in 5 animals). (**O–P**) Plot of normalized firing rates evoked by effective tones (at 60 dB SPL) with vs. without LP silencing (**O**) or LP activation (**P**) for the example cells shown above. Green and blue lines are the best fit linear regression line. Gray dashed line is the identity line. (**Q–S**) Summary of parameters of linear fitting for all neurons in LP silencing (ArchT) and activation (ChR2) groups, respectively. $R^2$ (**Q**): 0.84 ± 0.09 (mean ± SD, n = 8 cells) vs. 0.88 ± 0.08 (n = 13 cells); slope (**R**): 1.05 ± 0.17 (not significantly different from 1, p=0.41, *Z*-test) vs. 0.94 ± 0.17 (not significantly different from 1, p=0.26, *Z*-test); y-intercept (**S**): 0.23 ± 0.11 (significantly >0, p<0.001, *Z*-test) vs. −0.19 ± 0.12 (significantly <0, p<0.001, *Z*-test); Bar = SEM.

The online version of this article includes the following source data and figure supplement(s) for figure 1:

**Source data 1.** Data for *Figure 1* and *Figure 1—figure supplements 1–4*.
**Figure supplement 1.** LED illumination had no effect on auditory response.
**Figure supplement 2.** LP activity manipulation did not change CF of TRF.
**Figure supplement 3.** Effects on BW20.
**Figure supplement 4.** No effects in L4 of A1.

response level by either green or blue LED light delivery (*Figure 1—figure supplement 1B and D*). In L4, we did not observe any changes of either spontaneous or evoked FR induced by optogenetic manipulations of LP (*Figure 1—figure supplement 4*). Therefore, the LP's modulatory effect may be specific to superficial layers.

## LP exerts a thresholding effect on A1 L2/3 responses

Comparing frequency tuning curves without and with LP manipulation (*Figure 1C and J*), it appears that LP activity just shifted the A1 frequency tuning curve up or down, without changing the tuning preference (i.e. the best frequency). To further elucidate the nature of LP modulation, we plotted firing rates evoked by effective tones (see Materials and methods) with versus without LP manipulation and then performed a linear regression analysis. As shown by the two example A1 neurons (the same as shown in *Figure 1B and I*), data points were distributed along a line which had a positive y-intercept for LP silencing (*Figure 1O*) but a negative y-intercept for LP activation (*Figure 1P*), supporting the notion that all the tone responses were elevated or reduced by a certain amount when LP activity was manipulated. We did a similar analysis for all the recorded neurons and found that the linearity was generally high ($R^2$ close to 1) (*Figure 1Q*). The slope of the best fit line was close to 1 for both LP silencing and activation (*Figure 1R*), suggesting that there was no change in the response gain (i.e. no evidence for a multiplicative effect). The y-intercept was all positive for LP silencing but negative for LP activation (*Figure 1S*). These results strongly suggest a subtractive-suppression-like modulation of A1 responses by LP activity, similar to a thresholding effect produced by increasing the level of background noise (*Liang et al., 2014*). Therefore, LP can bidirectionally modulate auditory responses in superficial layers of A1 through a thresholding mechanism, that is, by shifting A1 frequency tuning up or down.

## LP modulation of A1 is mainly mediated by the LP-A1 projection

LP may modulate A1 responses through the direct LP to A1 projection, or indirectly through the LP projections to secondary cortices (*Arend et al., 2008*; *Cappe et al., 2009*; *Hackett et al., 1998*; *De La Mothe et al., 2006*; *Nakamura et al., 2015*; *Oh et al., 2014*). We wondered whether the direct LP-A1 projection contributed mainly to the observed modulatory effect of LP. By injecting a retrograde dye, CTB-488, in A1, we examined the distribution of LP neurons projecting to A1 (*Figure 2A*). Numerous retrogradely labeled neurons were found in LP (*Figure 2A*, right), with a bias towards its caudal part (*Figure 2—figure supplement 1*). Furthermore, injection of AAV-GFP into LP revealed densely labeled axon terminals in L1 as well as deep layers of A1 (*Figure 2B*), confirming the direct projection from LP to A1 (*De La Mothe et al., 2006*; *Nakamura et al., 2015*). We also observed dense LP projections to L4 of secondary auditory cortices (data not shown). To test whether the LP-A1 projection could account for the LP modulatory effect in A1, we optogenetically silenced the LP-A1 axon terminals by injecting AAV-eNpHR3.0 in LP and placing an optic fiber on the exposed A1 surface (*Figure 2C*). We observed effects on A1 L2/3 neurons similar to silencing LP neurons per se: the spontaneous and evoked FR was increased, the frequency tuning bandwidth was broadened, and the SNR was reduced (*Figure 2D–G*; *Figure 2—figure supplement 2A*). Conversely, we optogenetically activated the LP-A1 axon terminals by injecting AAV-ChR2 in LP and

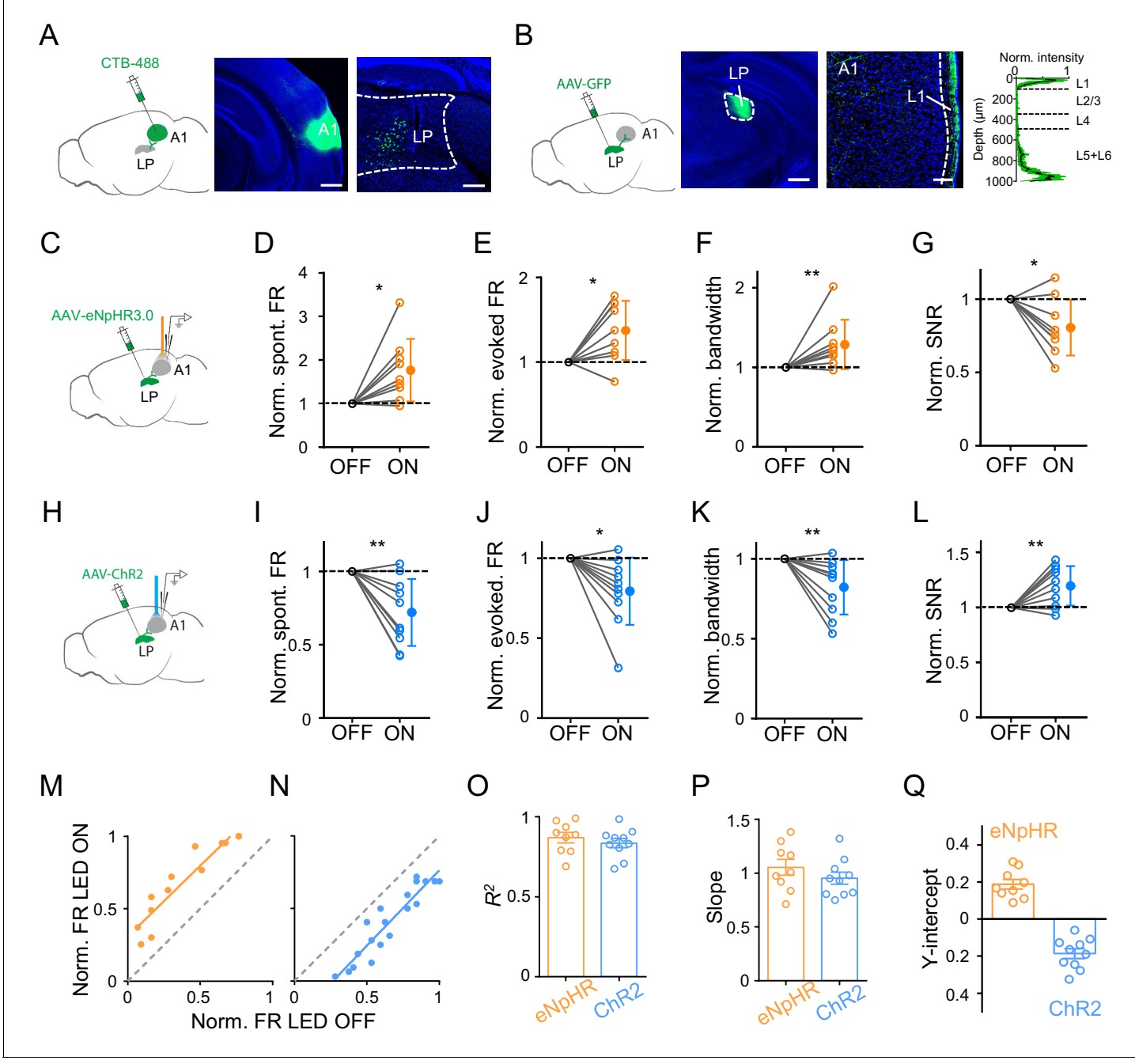

**Figure 2.** Bidirectional activity manipulations of the LP-A1 projection. (**A**) Left, injection of CTB-488 into A1. Middle, image showing fluorescence at the injection site. Scale bar, 500 µm. Right, retrogradely labeled neurons in LP. Scale bars, 200 µm. (**B**) Left, injection of AAV-GFP into LP. Middle, expression of GFP at the injection site. Scale bar, 500 µm. Right, GFP-labeled LP axons in A1 and average normalized fluorescence intensity across cortical depths (n = 6 brain sections, black line and green shade represent mean ± SD respectively). Scale bar, 100 µm. Cortical layers are marked. (**C**) Injection of AAV-eNpHR3.0 into LP and optical silencing of LP-A1 axon terminals by placing the optic fiber over A1. (**D–G**) Normalized spontaneous firing rate (**D**, p=0.013, paired *t*-test, n = 9 cells in 4 animals), evoked firing rate (**E**, p=0.012, paired *t*-test), tuning bandwidth (**F**, p=0.0078, Wilcoxon signed-rank test), and SNR (**G**, p=0.015, paired *t*-test) of A1 L2/3 neurons without and with LP-A1 axon terminal silencing. **p<0.01, *p<0.05. (**H**) Injection of AAV-ChR2 into LP and optical activation of LP-A1 axon terminals. (**I–L**) Normalized spontaneous firing rate (**I**, p=0.0038, paired *t*-test, n = 10 cells in 4 animals), evoked firing rate (**J**, p=0.013, paired *t*-test), tuning bandwidth (**K**, p=0.009, paired *t*-test), and SNR (**L**, p=0.0081, paired *t*-test) of A1 L2/3 neurons without and with LP-A1 axons terminal activation. (**M–N**) Plot of normalized firing rates evoked by effective tones (at 60 dB SPL) with vs. without LP-A1 axon terminal silencing (**M**) or activation (**N**) for two example cells. (**O–Q**) Summary of parameters of linear fitting in terminal silencing (eNpHR3.0) and activation (ChR2) groups, respectively. $R^2$ (**O**): 0.87 ± 0.098 (n = 9 cells) vs. 0.84 ± 0.091 (n = 10 cells); slope (**P**): 1.05 ± 0.22 (not

*Figure 2 continued on next page*

*Figure 2 continued*

significantly different from 1, p=0.48, *Z*-test) vs. 0.95 ± 0.18 (not significantly different from 1, p=0.42, *Z*-test); y-intercept (**Q**): 0.19 ± 0.077 (significantly >0, p<0.001, *Z*-test) vs. −0.19 ± 0.083 (significantly <0, p<0.001, *Z*-test). Bar = SEM.

The online version of this article includes the following source data and figure supplement(s) for figure 2:

**Source data 1.** Data for *Figure 2*.
**Figure supplement 1.** Projection from LP to A1.
**Figure supplement 2.** Effects of LP-A1 terminal manipulations on the TRF.

shining blue light on the exposed A1 surface (*Figure 2H*). In line with the effects caused by LP activation, the activation of LP-A1 axon terminals reduced the spontaneous and evoked FR, sharpened frequency tuning, and enhanced the SNR in A1 L2/3 pyramidal neurons (*Figure 2I–L*; *Figure 2—figure supplement 2B*).

We performed similar linear fitting on tone-evoked responses with versus without terminal manipulations (*Figure 2M and N*). We observed a thresholding effect similar to manipulating LP neurons per se: the linearity was high (*Figure 2O*), the slope of the best fit line was close to 1 (*Figure 2P*), and the y-intercept was a positive value for the terminal silencing while a negative value for the terminal activation (*Figure 2Q*). These data suggest that the LP to A1 projection largely mediates the LP modulatory effect on A1 responses.

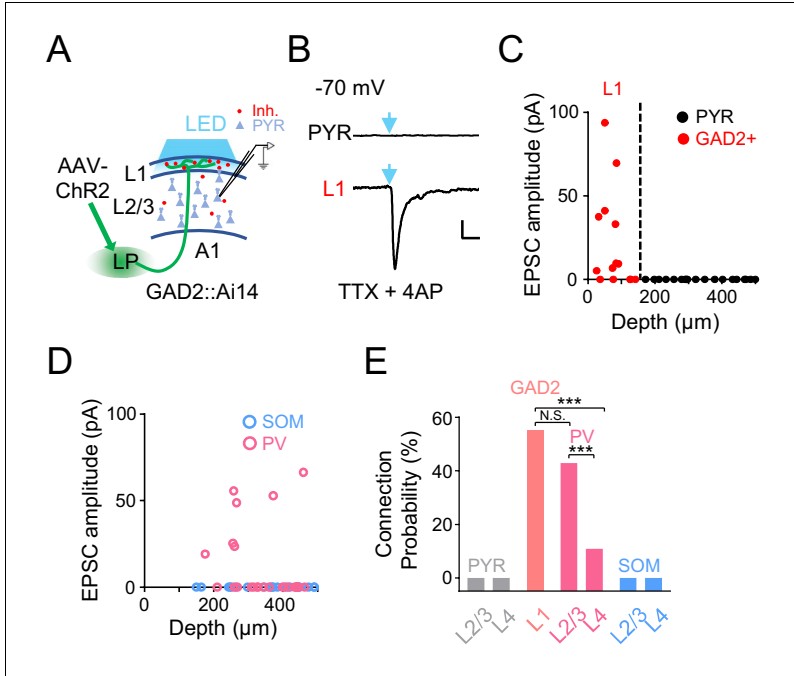

**Figure 3.** A1 cell types innervated by LP axons. (**A**) Slice recording paradigm. AAV-ChR2 was injected into LP. Blue LED light was applied to A1 to active LP-A1 axon terminals. Whole-cell recordings were made from A1 neurons. (**B**) Light-evoked monosynaptic EPSC (average trace) recorded (at −70 mV) in an example L2/3 pyramidal (PYR, top) and L1 inhibitory neuron (bottom). TTX and 4AP were present in the bath solution. Blue arrow marks the onset of 5 ms light pulse. Scale: 200 ms and 20 pA. (**C**) Plot of average amplitude of light-evoked monosynaptic EPSCs against the cell's cortical depth for all the recorded L1 inhibitory neurons (red, GAD2+) and pyramidal cells (black). Dashed line marks the boundary between L1 and L2/3. (**D**) Plot of average amplitude of light-evoked monosynaptic EPSCs against the cell's cortical depth for the recorded PV (magenta) and SOM (blue) neurons. (**E**) Summary of connection probability between LP-A1 axons and L1 GAD2+ neurons, as well as pyramidal, PV and SOM neurons in different layers. L1 vs. L2/3 PV, p=0.12; L1 vs. L4 PV, ***p<0.001; L2/3 PV vs. L4 PV, ***p<0.001, Fisher's exact test. N.S., not significant.

The online version of this article includes the following source data for figure 3:

**Source data 1.** Data for *Figure 3*.

## LP axons produce a disynaptic inhibitory effect on A1 L2/3 neurons

As the LP projection to the cortex per se is excitatory (*Evangelio et al., 2018*; *Roth et al., 2016*; *Zhou et al., 2018*), an immediate question is how LP activity exerts a net inhibitory effect on A1 L2/3 pyramidal neurons. Since LP axons project to L1 (*Figure 2B*), which contains predominantly inhibitory neurons (*Jiang et al., 2013*; *Schuman et al., 2019*), it is possible that LP axons can indirectly suppress L2/3 pyramidal neurons via L1 inhibitory neurons (*Ibrahim et al., 2016*; *Jiang et al., 2013*; *Zhou et al., 2014*). To test this idea, we injected AAV-ChR2 into LP and made whole-cell voltage-clamp recordings from A1 neurons in slice preparations (*Figure 3A*). Cuts were made in the tissue along boundaries between A1 and secondary auditory cortices to prevent potential feedback input to A1 (see Materials and methods). TTX and 4AP were present in the bath solution to ensure that only monosynaptic responses were recorded (*Petreanu et al., 2009*). We performed whole-cell recordings from several types of neurons in A1: L1 inhibitory neurons labeled by crossing GAD2-Cre with the Ai14 (Cre-dependent tdTomato) reporter, pyramidal neurons identified as tdTomato-negative cells in GAD2-Cre::Ai14 animals, parvalbumin (PV) and somatostatin (SOM) positive inhibitory neurons labeled by crossing PV-Cre or SOM-Cre with Ai14, respectively. As shown by two example cells, blue light activation of LP-A1 axons resulted in a robust excitatory postsynaptic current (EPSC) in the L1 inhibitory neuron, whereas no EPSC was observed in the pyramidal (PYR) cell (*Figure 3B*). Overall, none of the PYR neurons we recorded across L2-4 received direct input from LP, whereas more than 50% of L1 inhibitory neurons received direct LP input (*Figure 3C and E*). A similar fraction of PV neurons in L2/3 received direct input from LP, whereas none of the recorded SOM neurons did so (*Figure 3D and E*). A much smaller fraction of PV neurons in L4 also received LP input (*Figure 3E*). Together, these results indicate that LP-A1 axons preferentially innervate L1 inhibitory neurons and superficial-layer PV inhibitory neurons, which may then provide disynaptic inhibition to L2/3 pyramidal neurons.

## LP plays a role in noise-related contextual modulation of A1 responses

Previously, it has been proposed that LP can provide contextual information to visual cortex (*Fang et al., 2020*; *Roth et al., 2016*). Whether LP could play a similar role in auditory processing has been unknown. In an acoustic environment, one common contextual factor is the background noise. It has been shown that elevating the background noise level results in narrowing of frequency tuning of A1 neurons through a thresholding effect without changing the tuning preference (*Liang et al., 2014*). Since LP manipulations also produce a thresholding effect, we wondered whether LP could contribute to the noise-related contextual modulation. We noticed that LP neurons responded robustly to white-noise sound, with the response amplitude increasing with increasing noise levels (*Figure 4A*). Previous anatomical and electrophysiological studies have shown that the superior colliculus (SC) in the midbrain innervates LP (*Beltramo and Scanziani, 2019*; *Bennett et al., 2019*; *Fang et al., 2020*; *Gale and Murphy, 2014*; *Stepniewska et al., 2000*; *Wei et al., 2015*; *Zingg et al., 2017*). We thus wondered whether these noise responses in LP could be driven by SC input. Comparing the onset latency of noise responses in SC, LP, and A1 L4, we found that it was the shortest in SC, while similar between LP and A1 L4 (*Figure 4B*). This suggests that the responses in LP (at least the early part) are unlikely due to feedback inputs from auditory cortices. Furthermore, silencing SC greatly reduced the amplitude of noise responses in LP (*Figure 4C*). Our results thus indicate that the noise responses in LP are primarily driven by bottom-up input likely from SC.

To test whether the noise-driven LP activity affects A1 frequency processing, we applied tones (of varying frequencies) embedded in broadband noise of different levels (*Figure 4D and E*). Frequency tuning of A1 neurons was compared before and after silencing LP with bupivacaine (*Fang et al., 2020*; *Lee et al., 2008*; *Moraga-Amaro et al., 2014*). As shown by an example A1 L2/3 neuron (*Figure 4D and E*, gray), increasing the noise level reduced the amplitude of the response evoked by the best-frequency tone. Silencing LP resulted in a general increase in the tone-evoked response, regardless of the noise level, but to different degrees (*Figure 4D and E*, red). Summarizing all the recorded L2/3 neurons, we found that silencing LP universally enhanced tone-evoked responses in A1 across different noise levels (*Figure 4F*, upper panel). Notably, the enhancement was larger at higher noise levels (*Figure 4F*, lower panel). This is consistent with the notion that LP neuron responses increase with increasing noise levels (*Figure 4A*) and therefore exert a larger suppressive effect in A1 under a higher noise background. Silencing LP also elevated the spontaneous FR in a

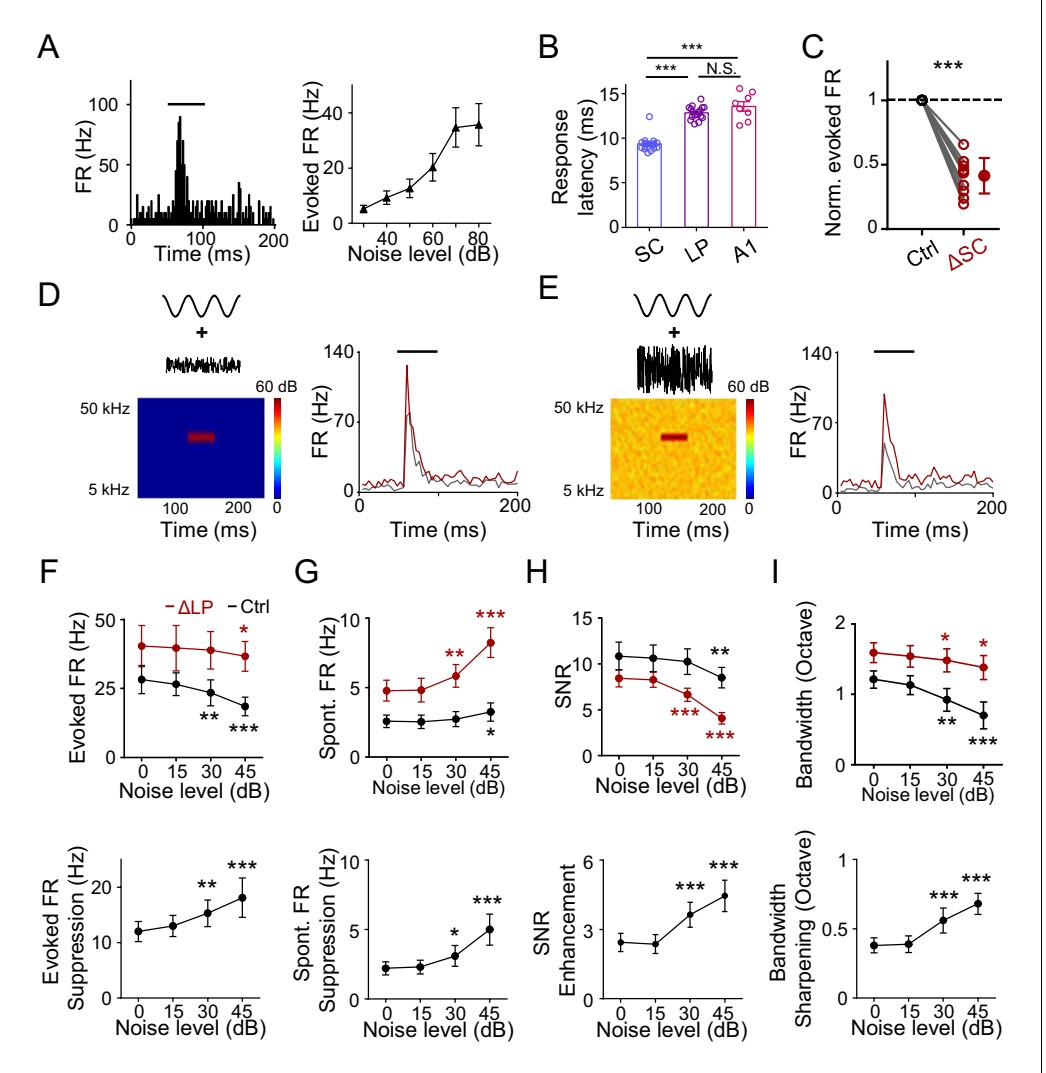

**Figure 4.** LP plays a role in noise-related contextual modulation of A1 responses. (**A**) Left, PSTH for responses of an example neuron in the caudal LP to white noise sound (marked by black line). Right, mean noise-evoked firing rate plotted against noise intensity (n = 10 LP neurons). Bar = SD. (**B**) Left, onset latency of noise (at 60 dB SPL) evoked responses in SC (n = 17 cells), LP (n = 17) and A1 L4 (n = 8) neurons. ***p<0.001, One-way ANOVA with Bonferroni's multiple comparisons test. N.S., not significant. (**C**) Normalized noise-evoked firing rate of LP neurons before (Ctrl) and after silencing SC (ΔSC). ***p<0.001, paired *t*-test, n = 12 cells in 2 animals. (**D**) Left, spectrogram of the stimulus: a 50 ms CF tone (at 60 dB SPL) embedded in low-level noise (at 0 dB SPL, 250 ms duration). Right, PSTHs for responses of an example A1 L2/3 neuron to the tone embedded in noise before (black) and after (red) silencing LP with bupivacaine. Black line marks the tone duration. (**E**) Response of the same cell to the same tone (60 dB SPL) embedded in higher-level noise (45 dB SPL) before and after silencing LP. (**F**) Upper, summary of evoked firing rates of A1 neurons at different noise levels before (black) and after (red) silencing LP. Lower, change in evoked firing rate by LP silencing at different noise levels. ***p<0.001, **p<0.01, *p<0.05, paired *t*-test, compared to the values under 0 dB noise condition, n = 9 cells from 4 animals. Bar = SD. (**G**) Summary of spontaneous firing rates before and after silencing LP. (**H**) Summary of SNR before and after silencing LP. (**I**) Summary of tuning bandwidths before and after silencing LP.

The online version of this article includes the following source data for figure 4:

**Source data 1.** Data for *Figure 4*.

noise-level dependent manner so that the elevation was larger at higher noise levels (*Figure 4G*). The SNR slightly decreased with increasing noise levels in the control condition (*Figure 4H*, upper panel, black), indicating that high-level noise has a detrimental effect on SNR, thus deteriorating auditory processing. Silencing LP not only reduced SNR, but also accelerated the detrimental effect of background noise (*Figure 4H*, upper panel, red). Again, the modulatory effect on SNR was larger at higher noise levels (*Figure 4H*, lower panel). Finally, the frequency tuning bandwidth was reduced with increasing noise levels (*Figure 4I*, upper panel), consistent with previous studies (*Ehret and Schreiner, 2000*; *Liang et al., 2014*). Silencing LP not only broadened the tuning bandwidth, but also slowed down the modulation of tuning bandwidth by increasing the noise level (*Figure 4I*). Together, these results suggest that LP plays a role in contextual modulation of A1 frequency processing by noise background.

## SC can drive the LP-mediated modulation of A1 responses

Since SC innervates LP (*Beltramo and Scanziani, 2019*; *Bennett et al., 2019*; *Gale and Murphy, 2014*; *Stepniewska et al., 2000*; *Wei et al., 2015*; *Zingg et al., 2017*) but does not project to auditory cortices (*Basso and May, 2017*; *Cang et al., 2018*; *Ito and Feldheim, 2018*), we wondered whether SC could provide direct input to drive the LP-mediated modulation of A1 responses. To confirm the connectivity from SC to LP, we injected AAV-GFP into intermediate and deep layers, the auditory related part of SC (*Bednárová et al., 2018*; *Drager and Hubel, 1975*; *King and Palmer, 1985*; *Meredith and Stein, 1986*; *Wise and Irvine, 1983*; *Zingg et al., 2017*). We found abundant GFP-labeled axons in LP (*Figure 5A*), with a strong bias towards its caudal part (*Figure 5—figure supplement 1*). Expressing ChR2 in intermediate and deep layers of SC and then performing whole-cell slice recording from caudal LP neurons in the presence of TTX and 4AP further confirmed direct innervations of LP neurons by SC axons (*Figure 5B and C*).

We next expressed ChR2 in intermediate and deep layers of SC (*Figure 5D*). Optogenetic activation of these SC neurons induced effects on A1 L2/3 pyramidal neurons similar to activation of LP: spontaneous and evoked firing rates were decreased, and SNR was increased (*Figure 5E–G*). We also expressed ChR2 in SC-recipient LP neurons (*Zingg et al., 2017*) by first injection of AAV-Cre in SC and second injection of AAV encoding Cre-dependent ChR2 in the caudal LP (*Figure 5H*). In A1, optogenetic activation of SC-recipient LP neurons produced similar effects to the activation of the general LP population: spontaneous and evoked firing rates were reduced, and SNR was increased (*Figure 5I–K*). Conversely, we silenced SC by infusing bupivacaine (*Figure 5L*). This resulted in increases of spontaneous and evoked firing rates, broadening of frequency tuning and a reduction of SNR in A1 L2/3 pyramidal neurons (*Figure 5M–P*), similar to silencing LP per se. Together, these results provide supporting evidence that SC can provide input to drive LP-mediated modulation of A1 responses.

## LP mediates looming visual stimuli induced modulation of auditory responses in A1

Both SC and LP process visual information and have been implicated in visual looming stimuli induced defensive behaviors such as freezing (*Wei et al., 2015*; *Yilmaz and Meister, 2013*; *Zingg et al., 2017*). Directly stimulating superficial-layer SC neurons, which receive visual input (*Zhao et al., 2014*; *Zhou et al., 2017*), can elicit freezing responses in mice (*Zingg et al., 2017*). Notably, LP neurons responded more strongly to visual looming stimuli compared to more commonly used grating and noise-pattern stimuli (*Figure 6—figure supplement 1*). These results imply that visual looming stimuli might be able to modulate A1 auditory responses via the SC-LP pathway. To test this idea, we paired sound stimulus with visual looming stimulus (*Figure 6A*), which was an expanding dark disk presented from the upper visual field (see Materials and methods), and compared auditory responses of A1 L2/3 pyramidal neurons with and without coupling the visual stimuli. In the presence of visual looming stimuli, spontaneous and evoked firing rates were reduced (*Figure 6B and C*), while the SNR was increased (*Figure 6D*). These results indicate that visual looming stimuli can indeed modulate A1 responses. After silencing LP with muscimol (*Figure 6E*), the effects on A1 responses by visual looming stimuli disappeared (*Figure 6F–H*), suggesting that LP primarily mediates the visual looming stimuli induced modulation of auditory responses in A1.

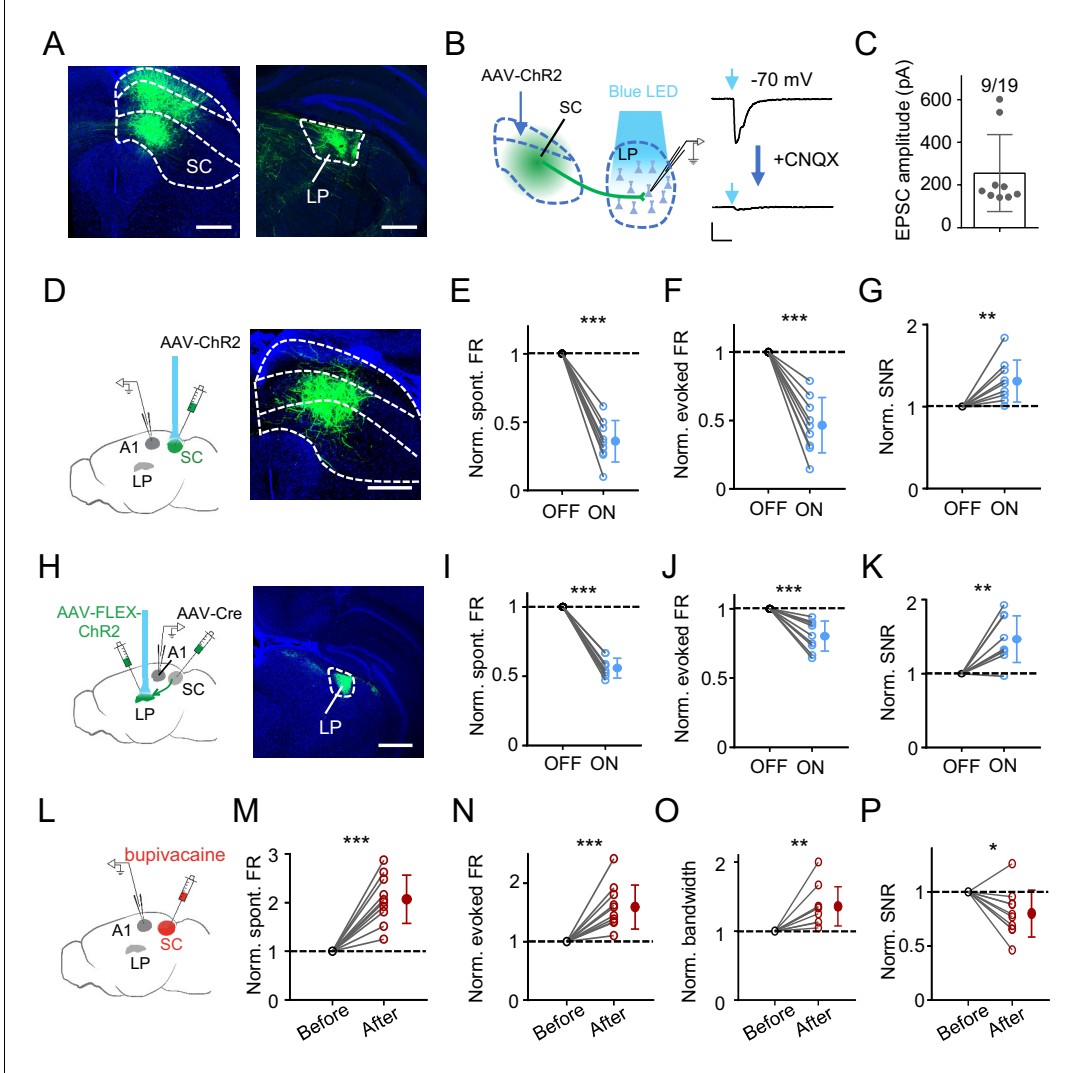

**Figure 5.** SC can drive LP-mediated modulation of A1 responses. (**A**) Injection of AAV-GFP into intermediate and deep layers of SC. Left, expression of GFP in SC. Boundaries between superficial, intermediate and deep layers are marked by dashed curves. Right, GFP labeled SC axons within LP. Scale bar, 500 μm. (**B**) Left, slice recording paradigm: expressing ChR2 in SC and whole-cell recording from LP neurons. Right, light-evoked monosynaptic EPSC (average trace) in an example LP neuron before and after perfusing in CNQX. Scale: 100 pA and 50 ms. (**C**) Average amplitudes of light-evoked monosynaptic EPSCs in LP neurons. Neurons not showing a light-evoked EPSC were excluded. (**D**) Left, illustration of optic activation of SC and recording in A1. Right, expression of ChR2 in SC. Scale bar, 500 μm. (**E–G**) Normalized spontaneous FR (**E**), evoked FR (**F**), and SNR (**G**) of A1 neurons without and with optical SC activation. ***p<0.001, **p=0.0065, paired *t*-test, n = 9 cells in 4 animals. (**H**) Transsynaptic labeling of SC-recipient LP neurons by first injection of AAV-Cre into intermediate and deep layers of SC and second injection of Cre-dependent ChR2 virus in LP. Right, expression of ChR2-EYFP in LP. Scale bar, 500 μm. (**I–K**) Normalized spontaneous FR (**I**), evoked FR (**J**), and SNR (**K**) of A1 L2/3 neurons without and with optical activation of SC-recipient LP neurons. ***p<0.001, **p=0.0022, paired *t*-test, n = 9 cells in 5 animals. (**L**) Experimental paradigm: silencing SC by infusing bupivacaine and recording in A1. (**M–P**) Normalized spontaneous FR (**M**), evoked FR (**N**), tuning bandwidth at 60 dB SPL (**O**) and SNR (**P**) of A1 L2/3 neurons before and after SC silencing. ***p<0.001, **p=0.0030, *p=0.017, paired *t*-test, n = 10 cells in 7 animals.

The online version of this article includes the following source data and figure supplement(s) for figure 5:

**Source data 1.** Data for *Figure 5*.
**Figure supplement 1.** SC projection to LP.

## Discussion

LP is considered a higher-order thalamic nucleus. In general, the influence of higher-order thalamus on auditory cortical processing has remained obscure. In the present study, our results demonstrate that LP activity can modulate auditory processing in superficial layers of A1. The overall outcome of

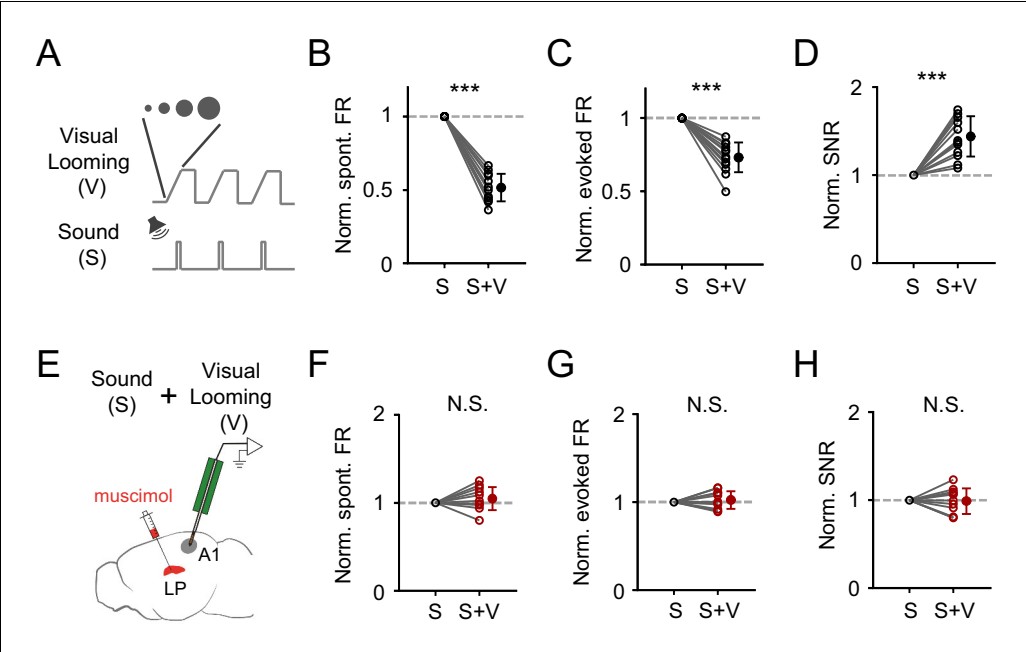

**Figure 6.** Visual looming stimuli modulate A1 auditory responses via LP. (**A**) Pairing of visual looming (V) and sound (S) stimulation. (**B–D**) Normalized spontaneous FR (**B**), evoked FR (**C**), and SNR (**D**) of A1 L2/3 neurons under sound only (S) and sound plus visual looming (S+V) condition. ***p<0.001, paired *t*-test, n = 12 cells from 2 animals. (**E**) Testing the effect of visual looming on A1 responses under condition of silencing LP with muscimol. (**F–H**) Normalized spontaneous FR (**E**, p=0.25, paired *t*-test, n = 12 cells from 2 animals), evoked FR (**F**, p=0.42, paired *t*-test, n = 12 cells), and SNR (**G**, p=0.79, paired *t*-test, n = 12 cells) of A1 neurons when LP was silenced. N. S., not significant.

The online version of this article includes the following source data and figure supplement(s) for figure 6:

**Source data 1.** Data for *Figure 6* and *Figure 6—figure supplement 1*.
**Figure supplement 1.** LP neurons are activated strongly by visual looming stimuli.

increasing LP activity is to improve auditory processing by sharpening frequency tuning and enhancing SNR of auditory evoked responses. This is achieved by subtractive suppression of auditory evoked responses together with suppression of spontaneous firing activity. We also demonstrate that such modulatory effect is largely mediated by the direct projection of LP to A1, where LP axons preferentially innervate L1 inhibitory neurons and superficial-layer PV inhibitory neurons, leading to a net disynaptic inhibitory effect on L2/3 pyramidal neurons. The suppression of A1 L2/3 responses is consistent with our recent study showing a similar suppressive effect in L2/3 of the primary visual cortex (V1), which results in an enhancement of visual feature selectivity (*Fang et al., 2020*).

The subtractive effect on L2/3 responses suggests that L1 inhibitory neurons (and PV neurons in superficial layers as well) may be broadly tuned for frequency. The frequency tuning property of L1 inhibitory neurons in the auditory cortex has not been studied yet, but a previous study in the visual cortex does suggest that L1 inhibitory neurons are broadly tuned for visual features such as orientation and direction (*Mesik et al., 2019*), and PV neurons in superficial layers of A1 have been shown to be broadly tuned for tone frequency (*Cohen and Mizrahi, 2015*; *Li et al., 2015*; *Liang et al., 2019*; *Maor et al., 2016*). As for the local circuit involved, we suggest that both L1 inhibitory neurons and L2/3 PV neurons, which receive direct LP input, can directly inhibit L2/3 pyramidal cells (*Jiang et al., 2013*), generating an overall suppressive effect. For L1 inhibitory neurons, their molecular identities have been reported to be diverse: 70% are NDNF+, 20% are nAchR+, and 10% are VIP+ (*Schuman et al., 2019*). It will be of great interest to investigate which of these cell types contributes to the LP-mediated modulation of A1 responses by using cell-type specific mouse lines. In addition, whether VIP+ neurons in L2/3 can also be involved in the suppressive effect remains to be tested.

The LP-mediated modulatory effect on A1 processing may be particularly pronounced and beneficial when there is a high noise background. A previous study has demonstrated that increasing background noise is equivalent to lowering the intensity of test (e.g. tone) stimuli, that is, shifting up the TRF by a certain Δthreshold value (*Liang et al., 2014*). In the current study, we demonstrate that LP activity also exerts a similar thresholding effect on A1 frequency tuning. Our results suggest that the noise effect on A1 frequency tuning may be achieved, at least partially, through LP. The contribution of LP to noise-related contextual modulation of A1 frequency processing is in a positive manner, in that it helps not only to prevent SNR of auditory responses from being deteriorated quickly by high-level noise background (*Figure 4H*) but also to accelerate the sharpening of frequency tuning with increasing noise levels (*Figure 4I*), which may compensate somewhat for the detrimental effect on SNR. Because LP activity is modulated by the noise level in that high-intensity noise activates LP neurons more than low-intensity noise (*Figure 4A*), LP neurons have a stronger protective effect on A1 processing under high-intensity than low-intensity noise conditions.

It is known that LP/pulvinar has extensive reciprocal connectivity with cortical areas including visual and auditory cortices (*Hackett et al., 1998*; *De La Mothe et al., 2006*; *Nakamura et al., 2015*; *Oh et al., 2014*; *Tohmi et al., 2014*; *Zhou et al., 2018*; *Bennett et al., 2019*). Previously, it has been suggested that LP/pulvinar serves in a cortico-thalamo-cortical ('transthalamic') indirect route for information transfer from one cortical area to another (*Guillery and Sherman, 2002*; *Sherman, 2016*). Besides cortical inputs, LP also receive strong inputs from SC (*Beltramo and Scanziani, 2019*; *Bennett et al., 2019*; *Fang et al., 2020*; *Gale and Murphy, 2014*; *Stepniewska et al., 2000*; *Wei et al., 2015*; *Zingg et al., 2017*). In this study, we demonstrate that LP (mainly the caudal part) receives direct input from the auditory related part of SC and projects to A1, and that silencing SC produces changes in A1 functional response properties similar to silencing LP. These results together suggest that SC can relay bottom-up input to LP to drive its modulation of A1 processing.

In addition, SC is a multisensory structure (*Drager and Hubel, 1975*; *King and Palmer, 1985*; *Meredith and Stein, 1986*). The visual only part of SC, that is the superficial layer of SC, also projects strongly to the caudal LP (*Beltramo and Scanziani, 2019*; *Bennett et al., 2019*; *Fang et al., 2020*). Thus, visual input may be able to modulate A1 responses as well via LP. Here, we demonstrate that visual looming stimuli modulate A1 L2/3 responses in a similar manner as increasing LP activity and that silencing LP blocks this modulation. This highlights a cross-modality feature of LP modulation. We should note however that the experiment with pharmacological silencing of LP (*Figure 6E–H*) does not exclude possible involvements of LP-mediated pathways other than the bottom-up SC-LP pathway. For example, visual looming signals may reach LP through feedback pathways from visual and associative cortices and then modulate responses in A1. We postulate that the multisensory nature of SC and its strong projections to LP potentially endow LP with an ability to modulate A1 activity given any salient sensory stimuli.

In summary, our results suggest that a previously unrecognized pathway, the SC-LP-A1 pathway, can provide contextual and cross-modality modulation of A1 responses and auditory processing to enhance the salience of acoustic information. How this pathway may interact with the canonical colliculo-thalamo-cortical auditory pathway remains to be investigated in the future.

## Materials and methods

**Key resources table**

| Reagent type (species) or resource | Designation | Source or reference | Identifiers | Additional information |
|---|---|---|---|---|
| Strain, strain background (*Mus musculus*) | C57BL/6J | Jachson Laboratory | RRID:IMSR_JAX:000664 | |
| Genetic reagent (*Mus musculus*) | Ai14 | Jachson Laboratory | RRID:IMSR_JAX:007914 | |
| Genetic reagent (*Mus musculus*) | GAD2-IRES-Cre | Jachson Laboratory | RRID:IMSR_JAX:010802 | |

*Continued on next page*

*Continued*

| Reagent type (species) or resource | Designation | Source or reference | Identifiers | Additional information |
|---|---|---|---|---|
| Genetic reagent (*Mus musculus*) | *PV-IRES-Cre* | Jachson Laboratory | RRID:IMSR_JAX:008069 | |
| Genetic reagent (*Mus musculus*) | *SOM-IRES-Cre* | Jachson Laboratory | RRID:IMSR_JAX:013044 | |
| Recombinant DNA reagent | AAV1-CAMKII-hChR2-eYFP | UPenn Vector Core | Addgene #26969 | |
| Recombinant DNA reagent | AAV1-CAG-ArchT-GFP | UPenn Vector Core | Addgene #29777 | |
| Recombinant DNA reagent | AAV1-hSyn-eNpHR3.0-mCherry | UPenn Vector Core | Addgene #26972 | |
| Recombinant DNA reagent | AAV2/1-CB7-Cl-eGFP-WPRE-rBG | UPenn Vector Core | Addgene #105542 | |
| Recombinant DNA reagent | AAV2/1-hSyn-Cre-WPRE-hGH | UPenn Vector Core | Addgene #105553 | |
| Recombinant DNA reagent | AAV2/1-EF1a-DIO-hChR2-eYFP | UPenn Vector Core | Addgene #105553 | |
| Chemical compound, drug | Cholera Toxin Subunit B (Recombinant), Alexa Fluor 488 Conjugate | ThermoFisher | C22841 | |
| Chemical compound, drug | NeuroTrace 640/660 Deep-Red Fluorescent Nissl Stain | ThermoFisher | N21483 | |
| Chemical compound, drug | Muscimol | ThermoFisher | M23400 | |
| Chemical compound, drug | Bupivacaine | Sigma-Aldrich | B1160000 | |
| Chemical compound, drug | Tetrodotoxin (TTX) | Torcris | Cat. No. 1078 | 1 µM |
| Chemical compound, drug | 4-Aminopyridine (4-AP) | Torcris | Cat. No. 0940 | 1 mM |
| Chemical compound, drug | Cyanquixaline (CNQX) | Sigma-Aldrich | C239 | 20 µM |
| Software, algorithm | Offline Sorter | Plexon | https://plexon.com | |
| Software, algorithm | MATLAB | Mathworks | https://www.mathworks.com/; RRID:SCR_001622 | |
| Software, algorithm | Prism | GraphPad | https://www.graphpad.com/scientific-software/prism/; RRID:SCR_00279 | |
| Software, algorithm | Fiji | NIH | https://fiji.sc/; RRID:SCR_002285 | |

## Experimental animals

All experimental procedures used in the present study were approved by the Animal Care and Use Committee at the University of Southern California. Male and female wild-type (C57BL/6J) and transgenic (GAD2-Cre, PV-Cre, SOM-Cre, and Ai14) mice were obtained from the Jackson Laboratory. Animals aged 8–12 weeks and weighed 18–28 g were used in the experiments. Animal sample sizes were determined by the estimated variances of the experiments and previous experience from similar experiments and were sufficient for all the statistical testes. Mice were housed under a 12 hr light/dark cycle. Food and water were provided ad libitum. Randomization methods were used to allocate experimental groups.

## Viral and neural tracer injection

Viral injections were performed as previously described *Fang et al. (2020)*; *Zingg et al. (2017)*. In brief, stereotaxic coordinates were selected based on the Allen Mouse Brain Atlas (www.brain-map. org). Mice were placed on a heating pad with homoeothermic control and anesthetized with 1.5% isoflurane throughout all surgical procedures. A small cut was made on the skin after shaving to expose the skull. A craniotomy of 0.2 mm in diameter was made to expose the underlying cortex. After removing the dura mater, a pulled glass pipette with a beveled tip of ~20 μm in diameter was inserted to the target region. The viral solution or tracers were delivered by pressure injection. For optogenetic silencing and activating LP and its terminals, AAV1-CAMKII-hChR2-eYFP (UPenn Vector Core, $1.7 \times 10^{13}$ GC/ml), AAV1-CAG-ArchT-GFP (UPenn Vector Core, $1.7 \times 10^{13}$ GC/ml), and AAV1-hSyn-eNpHR3.0-mCherry (UPenn Vector Core, $1.7 \times 10^{13}$ GC/ml) was injected into LP (50 nl total volume, AP −2.4 mm, ML +1.6 mm, DV −2.4 mm) of wild-type animals, respectively. For anterograde tracing of LP projections, AAV2/1-CB7-Cl-eGFP-WPRE-rBG (UPenn Vector Core, $1.7 \times 10^{13}$ GC/ml) was injected into LP. For retrograde tracing of LP afferents, we injected CTB488 into A1 (AP −2.6 mm, ML +4.4 mm, DV +0.6 mm). For optogenetically activating SC axonal terminals in LP for slice recording, AAV1-CAMKII-hChR2-eYFP (UPenn Vector Core, $1.7 \times 10^{13}$ GC/ml) was injected into SC (AP −3.75 mm, ML +0.6 mm, DV −1.45 mm). For transsynaptic labeling from SC to LP, AAV2/1-hSyn-Cre-WPRE-hGH (UPenn Vector Core, $2.5 \times 10^{13}$ GC/mL) was injected into SC, and AAV2/1-EF1a-DIO-hChR2-eYFP (UPenn Vector Core, $1.6 \times 10^{13}$ GC/mL) was injected into LP. Animals were allowed to recover for at least 3 weeks following the injections of viruses.

## Histology and imaging

After experiments, animals were deeply anesthetized and transcardially perfused with phosphate buffered saline (PBS) followed by 4% paraformaldehyde (PFA). The brain tissue was collected and fixed in 4% PFA at 4 ˚C overnight and then sliced into 150 μm sections with a vibratome (Leica, VT1000s). Nissl staining was used to visualize the cyto-architecture. Brain slices were first rinsed with PBS for 10 min, and then incubated in PBS containing Neurotrace 620 (ThermoFisher, N21483) and 0.1% Triton X-100 (Sigma-Aldrich) for 2 hr. Images were taken with a confocal microscope (Olympus FluoView FV1000). To identify and verify the injection site and spread of virus expression and drug infusion, images were taken under a 4 × objective. Regions with axonal labeling were further imaged under a 10 × objective for clearer visualization of innervation patterns.

## Optogenetic and pharmacological manipulation

One week before recording sessions, the animals were prepared as previously described *Chou et al. (2018)*; *Fang et al., 2020*. In brief, mice were placed on the stereotaxic apparatus and anesthetized with 1.5% isoflurane during the implantation. Optic fiber implantation was made at least three weeks after injecting ChR2 or ArchT virus. For implantation, small holes of 500 μm in diameter were drilled to allow the insertion of microinjection tubes (300 μm ID, RWD) or the fiber optic cannula (200 μm ID, Thorlabs). The holes were drilled directly above LP (AP −2.5 mm, ML +1.6 mm), or A1 (AP −2.6 mm, ML +4.4 mm). The injection tube or cannula were lowered to the desired depth and fixed in place using dental cement. In the meantime, a screw for head fixation was mounted on top of the skull with dental cement as well. For drug infusion during recording, an injector (100 μm ID, RWD) was inserted into the microinjection tube, and 200 nl 0.5% bupivacaine mixed with DiI (2 mg/ml) or fluorescent muscimol (1.5 mM, Life Technologies) were slowly injected into SC (AP −3.75 mm, ML +0.6 mm, DV −1.45 mm) or LP (AP −2.5 mm, ML +1.6 mm, DV −2.4 mm)

through a micro-syringe. Since bupivacaine's silencing effect lasts for only 30–40 min (*Fang et al., 2020*), it was possible for us to sequentially record from multiple A1 neurons in the same animal and examine effects on them by silencing LP/SC. Optogenetic manipulations were performed following our previous studies (*Chou et al., 2018*; *Fang et al., 2020*; *Ibrahim et al., 2016*). To optogenetically activate LP, light from a blue LED source (470 nm, 10 mW, Thorlabs) was delivered at a rate of 20 Hz (20 ms pulse duration) via the implanted cannulas using a patch cord (Ø200 μm, 0.22 NA SMA 905, Thorlabs) for ChR2 animals. The plastic sleeve (Thorlabs) securing the patch cord and cannula was wrapped with black tape to prevent light leakage. Similarly, for silencing LP, light from a green LED source (530 nm, 10 mW, Thorlabs) was delivered continuously for stimulating ArchT-expressing neurons. For manipulation of terminals from LP to A1, optic cannula connected with a blue and an amber LED (589 nm, 10 mW, Thorlabs) source was placed on the surface of A1 to stimulate ChR2- and eNpHR3.0-expressing LP axons, respectively. The optogenetic stimulation preceded sound stimulation by 50 ms. Animals were allowed to recover for one week before recording session. During the recovery period, they were habituated to the head fixation on the flat running plate. The head screw was tightly fit into a metal post while the animal could run freely on the plate. Following recording sessions, animals were euthanized, and the brain was imaged to verify the specificity of virus expression and locations of implantations. Mice with mistargeted injections or misplacements of drug infusion tubes or optic fibers were excluded from data analysis.

## Slice preparation and recording

Three weeks following the injections, animals were decapitated following urethane anesthesia and the brain was rapidly removed and immersed in an ice-cold dissection buffer (composition: 60 mM NaCl, 3 mM KCl, 1.25 mM NaH$_2$PO$_4$, 25 mM NaHCO$_3$, 115 mM sucrose, 10 mM glucose, 7 mM MgCl$_2$, 0.5 mM CaCl$_2$; saturated with 95% O$_2$ and 5% CO$_2$; pH = 7.4). Coronal slices at 350 μm thickness were sectioned by a vibrating microtome (Leica VT1000s), and recovered for 30 min in a submersion chamber filled with warmed (35°C) ACSF (composition:119 mM NaCl, 26.2 mM NaHCO$_3$, 11 mM glucose, 2.5 mM KCl, 2 mM CaCl$_2$, 2 mM MgCl$_2$, and 1.2 mM NaH$_2$PO$_4$, 2 mM Sodium Pyruvate, 0.5 mM Vitamin C). A1 or LP neurons were visualized under a fluorescence microscope (Olympus BX51 WI). Patch pipettes (~4–5 MΩ resistance) filled with a cesium-based internal solution (composition: 125 mM cesium gluconate, 5 mM TEA-Cl, 2 mM NaCl, 2 mM CsCl, 10 mM HEPES, 10 mM EGTA, 4 mM ATP, 0.3 mM GTP, and 10 mM phosphocreatine; pH = 7.25; 290 mOsm) were used for whole-cell recordings. Signals were recorded with an Axopatch 700B amplifier (Molecular Devices) under voltage clamp mode at a holding voltage of –70 mV for excitatory currents or 0 mV for inhibitory currents, filtered at 2 kHz and sampled at 10 kHz. Tetrodotoxin (TTX, 1 μM) and 4-aminopyridine (4-AP, 1 mM) were added to the external solution to isolate monosynaptic responses. Blue light stimulation (10 ms pulse, 3 mW power, 10–30 trials) was delivered via a mercury Arc lamp gated with an electronic shutter. PV+, SOM+ or L1 GAD2+ neuronal types were determined by the tdTomato expression. Laminar location of the recorded cell was determined by its depth from the pial surface. CNQX (10 μM) was added to verify glutamatergic transmission.

## Sound and visual stimulation

White noise (50 ms, 70 dB sound pressure level or SPL) or tone pips (50 ms duration, 3 ms ramp) of various frequencies (2–45.25 kHz, 0.1 octave interval) and intensities (10–70 dB SPL, at 10 dB interval) were generated by custom-made software in LabView (National Instruments) through a 16-bit National Instruments interface, and delivered through a calibrated speaker (Tucker-Davis Technologies) to the contralateral ear. The 322 testing stimuli were presented in a pseudorandom sequence. The inter-stimulus interval between noise stimulation or tone pips was 1 s. For auditory stimuli embedded in noise of different levels, 50 ms tone pips at different intensities and frequencies as described above were embedded in white noise of different intensity levels (0, 15, 30, 45 dB SPL). The white noise was 250 ms in duration and preceded the tone pip by 100 ms.

Visual stimuli were generated in Matlab (MATLAB) with the Psychophysics Toolbox Version 2 (*Brainard, 1997*) and were presented on a ViewSonic VA705b monitor (1920 × 1440 pixels, 33.9 cm wide, 27.2 cm high, 60 Hz refresh rate, mean luminance 41 cd/m$^2$) mounted on a flexible arm. For visual noise stimulation, a set of dense white-noise patterns were presented. Each frame of the stimuli consisted of a grid of 20 × 20 squares (each square 4°×3°), intensities of which were determined

by a m-sequence. Each pattern was presented for 200 ms and 30–50 patterns were presented according to the response level and fidelity. For grating stimulation, drifting sinusoidal gratings (12 directions, 0°−330°, 30° per step, 6 orientations) were presented in a pseudorandom order for 5–8 repetitions. The spatial frequency of the gratings was chosen to be 0.04 cycles per second (cpd) and temporal frequency to be 2 cycles per second (Hz). Each grating drifted for 1.5 s and another grating appeared, which remained to be static for 3 s before drifting. The looming stimulus was an expanding black disk presented from the upper visual field. It changed from 2° to 20° size (in diameter) within 250 ms. The largest black disk after the expansion stayed for another 250 ms, and the interstimulus interval was 250 ms. The auditory stimulus (tone pips as described above) was paired with each cycle of the looming stimulation with an onset delay of 200 ms. We usually only presented 30 cycles of looming stimulation to avoid a potential adaptation effect.

## In vivo electrophysiology

One week after the preparation, animals were head-fixed on the running plate, and electrophysiology recording with either optogenetic or pharmacological manipulation was carried out in a sound-attenuation booth. Loose-patch recordings were performed as previously described (*Fang et al., 2020*; *Ibrahim et al., 2016*; *Liang et al., 2019*; *Zhou et al., 2014*), with a patch pipette filled with an artificial cerebral spinal fluid (ACSF; 126 mM NaCl, 2.5 mM KCl, 1.25 mM $Na_2PO_4$, 26 mM $NaHCO_3$, 1 mM $MgCl_2$, 2 mM $CaCl_2$ and 10 mM glucose). A loose seal (0.1–0.5 GΩ) was made on the cell body, allowing spikes only from the patched cell to be recorded. Signals were recorded with an Axopatch 200B amplifier (Molecular Devices) under voltage-clamp mode, with a command voltage applied to adjust the baseline current to near zero. Loose-patch recording signals were filtered with a 100–5,000 Hz band-pass filter. The depths of the recorded neurons were determined based on the micromanipulator reading. Spikes could be detected without ambiguity because their amplitudes were normally higher than 100 pA, while the baseline fluctuation was <5 pA. Multichannel recordings were carried out by lowering a 64-channel silicone probe (NeuroNexus) into the target region. Signals were recorded by an Open-Ephys system. Multi-unit signals during sound stimulation were recorded and saved for offline analysis.

## Data analysis

Noise-driven and tone-driven spike rates were analyzed within a 10–60 ms time window after the onset of sounds. For quantifying evoked firing rates, average baseline firing rate calculated within a 50 ms time window preceding the onset of sounds was subtracted. For quantifying changes in spontaneous firing rate, spontaneous firing rates calculated within a 50 ms time window before sound onsets were compared between LED-on and LED-off conditions. TRFs were reconstructed according to the array sequence. The frequency–intensity space was up-sampled 5 times along the frequency and intensity dimensions only for visualization purposes. The measurements of TRF parameters (e.g. bandwidth) were made from the raw data. Boundaries of the spike TRF were determined following previous studies (*Liang et al., 2014*; *Schumacher et al., 2011*; *Sutter and Schreiner, 1991*; *Xiong et al., 2013*). A threshold at the value equal to the spontaneous spike rate plus 20% of the peak evoked firing rate was then used to define significant evoked responses. Responses to frequency–intensity combinations that met this criterion were considered to fall within the TRF of the neuron, which generated the contour of the TRF (*Xiong et al., 2013*). Characteristic frequency (CF) was defined as the frequency (in Hz) at which the lowest sound pressure level was necessary to evoke a significant excitatory response. Bandwidth of TRF was determined as the total frequency range for effective tones at 60 dB SPL or at 20 dB SPL above the intensity threshold (i.e. BW20). SNR of auditory responses was calculated as the evoked firing rate divided by the spontaneous firing rate. To quantify changes in evoked firing rate, average tone-evoked firing rates across the TRF (under the control condition) were compared between control and activity manipulation conditions. Onset latency of spike responses was determined based on the generated peristimulus spike-time histogram (PSTH) as the interval between the stimulus onset and the time point where spike rate exceeded the average baseline by 2 standard deviations of baseline fluctuations.

To plot frequency tuning curve using the Envelope function (*Sun et al., 2010*), the tone-evoked responses at 60 dB SPL with and without activity manipulation were normalized to the highest evoked firing rate in the control condition. Linear regression fitting for the normalized evoked firing

rates with and without activity manipulation was performed for individual neurons in each group, and $R^2$, slope, and intercepts of the best fit line were determined.

For spike sorting of data obtained from multichannel recording, spike signals were filtered with a 300–3,000 Hz band-pass filter. The nearby four channels of the silicon probe were grouped as tetrodes and semi-automatic spike sorting was performed using the offline sorter of Plexon (Dallas, Texas) following our previous study (*Zhang et al., 2018*). Clusters with isolation distance >20 were considered as separate clusters. Spike clusters were classified as single units only if the waveform SNR (signal-to-noise ratio) exceeded 4 (12 dB) and the inter-spike interval was longer than 1.2 ms for >99.5% of the spikes.

## Statistics

Shapiro–Wilk test was first applied to examine whether samples had a normal distribution, and a p value < 0.05 indicated non-normality. In the case of a normal distribution, *Z*-test or two-sided paired *t*-test was applied. Otherwise, the Wilcoxon signed-rank test was applied as a non-parametric test. Two-sample Kolmogorov-Smirnov test was used to test whether data from two groups were from the same distribution. Statistical analysis was conducted with Matlab. Data were reported as mean ± SD unless otherwise mentioned.

## Acknowledgements

This work was supported by grants from the US National Institutes of Health to LIZ (R01DC008983; RF1MH114112) and HWT (EY019049 and EY022478).

## Additional information

### Funding

| Funder | Grant reference number | Author |
| --- | --- | --- |
| National Institutes of Health | R01DC008983 | Li I Zhang |
| National Institutes of Health | RF1MH114112 | Li I Zhang |
| National Institutes of Health | EY019049 | Huizhong W Tao |
| National Institutes of Health | EY022478 | Huizhong W Tao |

The funders had no role in study design, data collection and interpretation, or the decision to submit the work for publication.

### Author contributions

Xiao-lin Chou, Qi Fang, Conceptualization, Data curation, Formal analysis, Investigation, Writing - original draft; Linqing Yan, Conceptualization, Data curation, Formal analysis, Investigation; Wen Zhong, Jinxing Wei, Data curation, Investigation; Bo Peng, Haifu Li, Data curation, Formal analysis, Investigation; Huizhong W Tao, Li I Zhang, Conceptualization, Supervision, Funding acquisition, Validation, Investigation, Visualization, Methodology, Project administration, Writing - review and editing

### Author ORCIDs

Huizhong W Tao  https://orcid.org/0000-0002-3660-0513
Li I Zhang  https://orcid.org/0000-0003-0275-8651

### Ethics

Animal experimentation: This study was performed in strict accordance with the recommendations in the Guide for the Care and Use of Laboratory Animals of the National Institutes of Health. All of the animals were handled according to approved institutional animal care and use committee (IACUC) protocol (protocol number: 21109) of the University of Southern California.

Decision letter and Author response
Decision letter https://doi.org/10.7554/eLife.54157.sa1
Author response https://doi.org/10.7554/eLife.54157.sa2

## Additional files

### Supplementary files
• Transparent reporting form

### Data availability
All data generated or analysed during this study are included in the manuscript and supporting files. Source data files have been provided for all figures.

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
