## [Decision Letter]

**Acceptance summary:**

This article demonstrates for the first time the role of the lateral posterior nucleus of the thalamus (LP), a part of the pulvinar, in controlling auditory cortex activity according to auditory but also visual context. Particularly interesting are the demonstrations that LP is critical for maintaining good signal-to-noise ratio for tone responses in the presence of background noise, and that it also improves auditory coding during visual threat, thanks to inputs received from the superior colliculus. This indicates non-trivial influences of secondary thalamic nuclei on auditory processing.

**Decision letter after peer review:**

Thank you for submitting your article "Contextual and Cross-Modality Modulation of Auditory Cortical Processing through Pulvinar Thalamus Mediated Thresholding" for consideration by *eLife*. Your article has been reviewed by three peer reviewers, including Brice Bathellier as the Reviewing Editor and Reviewer #1, and the evaluation has been overseen by Andrew King as the Senior Editor. The following individual involved in review of your submission has agreed to reveal their identity: Alfonso junior Apicella (Reviewer #3).

The reviewers have discussed the reviews with one another and the Reviewing Editor has drafted this decision to help you prepare a revised submission.

Summary:

The authors show that the analog of pulvinar in rodent, LP, projects to auditory cortex. Moreover its activation modulates spontaneous activity and auditory responses, putatively by driving GABAergic interneurons (in L1 but also PV cells in L2/3). This results in a decreased signal-to-noise ratio and sharper frequency tuning curves. Also, LP seems to be driven by superior colliculus and mediates inhibition of AC (but increases SNRs for sound response) during dark looming visual stimuli. Finally another interesting result is that LP activation improves SNRs in acoustic noise.

This is a really nice paper that studies the projection from LP to A1 in awake mice using a barrage of different circuit-specific tools. The authors elegantly demonstrate that the lateral posterior nucleus (LP) of the thalamus, the rodent homologue of the primate pulvinar thalamus, contributes to the maintenance and enhancement of primary auditory cortex processing in the presence of auditory background noise and threating visual looming stimuli, respectively. The study design is uncomplicated, and the experiments were carried out very carefully. The results are presented clearly and are overall convincing. Together, this makes an interesting study about a potentially ecologically relevant modulation of auditory cortex, in a multisensory context.

Nevertheless the reviewers pointed to a number of essential revisions to make in order to clarify particular points of the manuscript.

Essential revisions:

1) The authors claim throughout the paper that LP modulates auditory responses but that this modulation is purely additive and occurs both in spontaneous and evoked response. This seems to be more a global offset than a modulation. Unfortunately, we could not find in the text or Materials and methods section whether evoked firing rates are calculated after subtraction of baseline or not, but it seems it is done without baseline subtraction.

If this is correct, then baseline-subtracted evoked response would likely show no effect of LP (in)activation.

The authors should clarify that and mitigate their claim about a modulation while their main results rather describe an input bias acting both on evoked and spontaneous firing rates with the same strength.

Note that this 'input bias description' explains very well the SNR increase: SNR = (signal + LP_offset)/(baseline + LP_offset) as signal > baseline, LP_offset has a stronger impact on the denominator, and SNR increases for a negative offset.

If this is correct, it would be clearer to say that LP activity removes some of the background activity to sharpen the actual signal, but without modulating the signal itself. In any case, baseline subtracted responses should be added to the manuscript and it should be discussed if the observed modulation is a simple offset or more than that.

That said, the very interesting results in acoustic noise conditions show that LP can also have an effect on responses to acoustic noise, which does not seem to be purely additive, as in that case LP silencing has much more impact for loud than quiet white noise. In this specific case, one observes a real down-modulation of the noise response by LP. So this should also be mentioned and it should be discussed why LP effects are not additive anymore in this case.

Also the authors could discuss potential operative models explaining how interneuron activations lead to the observed modulations. For example, are the authors able to say anything about the L1 inhibitory tuning properties? i.e. what is the local circuit mechanism that is leading to the L2/3 changes? Do you know anything about the identity of the L1 neurons contributing to the circuit (are they primarily VIP+)?

2) It seems from the figures and the Materials and methods that the receptive fields were upsampled. Were the bandwidths (and anything else extracted from the receptive fields) taken from the upsampled data? If so, this renders the result dependent on the technique used for upsampling. In this case measurements such as bandwidth should be extracted from raw data.

3) Why was 60dB used for your bandwidth measurement. It seems a little arbitrary given that the standard in the auditory field is to use "BW10" or "BW20" (i.e. bandwidths either 10 or 20 dB above threshold). Please, use these measurements as well. In looking at Figure 1B one cannot help but think that 60 dB was chosen because lower intensities did not exhibit a significant result – is that the case?

4) Could the authors elaborate a little of their choice of light power for activation/inactivation experiments? From the Materials and methods, it seems like 10mW was used. Was the same power used for both activation/inactivation experiments (and is this valid, given the differences between ChR2 and Arch)? Did they titrate power at all, i.e. if the power was kept the same for all experiments, then day-to-day experimental variability (differences in viral uptake, variation in fiber placement) become an additional confound.

5) The statistics used in the paper are insufficiently documented, and could be improved. In the Materials and methods section, it is stated that Shapiro-Wilk was used as a normality test prior to a paired t-test, if appropriate. However the results are never shown and the paired t-test seems to have been used for the majority of tests between on and off conditions throughout the paper. Please justify systematically the normality of the data or even better, use a non-parametric test (e.g. Signed test), which should just work as well.

6) Things become a little bit more circuit unspecific later in the paper. For example, the visual looming experiment is only carried out with a mass pharmacological silencing of LP. The results are certainly suggestive, given the rest of the paper, but alternate circuits and mechanisms should be thoroughly discussed and acknowledged.

---

## [Author Response]

Essential revisions:1) The authors claim throughout the paper that LP modulates auditory responses but that this modulation is purely additive and occurs both in spontaneous and evoked response. This seems to be more a global offset than a modulation. Unfortunately, we could not find in the text or Materials and methods section whether evoked firing rates are calculated after subtraction of baseline or not, but it seems it is done without baseline subtraction.If this is correct, then baseline-subtracted evoked response would likely show no effect of LP (in)activation.The authors should clarify that and mitigate their claim about a modulation while their main results rather describe an input bias acting both on evoked and spontaneous firing rates with the same strength.Note that this 'input bias description' explains very well the SNR increase: SNR = (signal + LP_offset)/(baseline + LP_offset) as signal > baseline, LP_offset has a stronger impact on the denominator, and SNR increases for a negative offset.If this is correct, it would be clearer to say that LP activity removes some of the background activity to sharpen the actual signal, but without modulating the signal itself. In any case, baseline subtracted responses should be added to the manuscript and it should be discussed if the observed modulation is a simple offset or more than that.

We apologize for the inadequate description about data analysis in the previous manuscript. For quantifying evoked firing rates, we have subtracted the baseline firing rate, which was calculated within a 50-ms time window preceding the onset of sound stimulation. For quantifying changes in spontaneous firing rate, we compared the spontaneous firing rates calculated within a 50-ms window before sound stimulation between LED-On and LED-Off conditions (LED stimulation preceded sound stimulation by 50 ms). We have clarified these details in the Materials and methods (subsection “Sound and visual stimulation”). Thus, the changes in evoked firing rate we report here represent a bona fide modulation of signals separate from the effect on the baseline activity.

That said, the very interesting results in acoustic noise conditions show that LP can also have an effect on responses to acoustic noise, which does not seem to be purely additive, as in that case LP silencing has much more impact for loud than quiet white noise. In this specific case, one observes a real down-modulation of the noise response by LP. So this should also be mentioned and it should be discussed why LP effects are not additive anymore in this case.

That LP silencing has more impact for loud than quiet noise (Figure 4F) can be explained by the modulation of LP neuron activity per se by noise levels (see Figure 4A). Because loud noise activates LP neurons more strongly than soft noise, LP neuron activity has a stronger suppressive effect in A1 under the loud than soft noise condition. The additive effect we have described for frequency tuning changes only applies for a certain background noise condition, i.e. the background noise remains constant while tones of different frequencies are tested. We have clarified this point in the text (Discussion, third paragraph).

Also the authors could discuss potential operative models explaining how interneuron activations lead to the observed modulations. For example, are the authors able to say anything about the L1 inhibitory tuning properties? i.e. what is the local circuit mechanism that is leading to the L2/3 changes? Do you know anything about the identity of the L1 neurons contributing to the circuit (are they primarily VIP+)?

We thank the reviewers for raising these questions. The additive effect suggests that L1 inhibitory neurons (and L2/3 PV neurons as well) are very broadly tuned for frequency. There has not been any published study on the frequency tuning property of L1 inhibitory neurons in auditory cortex yet. However, a previous study in the visual cortex does suggests that L1 inhibitory neurons are broadly tuned for visual features such as orientation and direction (Mesik et al., 2019) and PV neurons in superficial layers of A1 have been shown to be broadly tuned for frequency (Cohen and Mizrahi, 2015; Li et al., 2015; Liang et al., 2018; Maor et al., 2016). As for the local circuit involved, we postulate that both L1 inhibitory neurons and L2/3 PV cells can directly inhibit L2/3 pyramidal neurons (Jiang et al., 2013), generating the overall suppressive effect.

The molecular identity of L1 neurons is diverse: 70% are NDNF+, 20% are nAchR+, and 10% are VIP+ (Schuman et al., 2019). We will need cell-type specific mouse lines to test which of these cells types contributes most, but it is unlikely that VIP+ neurons are the answer. We have added these points in the Discussion (second paragraph).

2) It seems from the figures and the Materials and methods that the receptive fields were upsampled. Were the bandwidths (and anything else extracted from the receptive fields) taken from the upsampled data? If so, this renders the result dependent on the technique used for upsampling. In this case measurements such as bandwidth should be extracted from raw data.

We would like to clarify that the receptive fields were upsampled only for visualization purposes. The measurements of receptive field parameters (e.g. bandwidth) were made from the raw data. We have clarified this point in the Materials and methods (subsection “Data analysis”).

3) Why was 60dB used for your bandwidth measurement. It seems a little arbitrary given that the standard in the auditory field is to use "BW10" or "BW20" (i.e. bandwidths either 10 or 20 dB above threshold). Please, use these measurements as well. In looking at Figure 1B one cannot help but think that 60 dB was chosen because lower intensities did not exhibit a significant result – is that the case?

As suggested by the reviewers, we have analyzed BW20. Our results show that the change in bandwidth is still significant (see new Figure 1—figure supplement 3).

4) Could the authors elaborate a little of their choice of light power for activation/inactivation experiments? From the Materials and methods, it seems like 10mW was used. Was the same power used for both activation/inactivation experiments (and is this valid, given the differences between ChR2 and Arch)? Did they titrate power at all, i.e. if the power was kept the same for all experiments, then day-to-day experimental variability (differences in viral uptake, variation in fiber placement) become an additional confound.

We used 10 mW power for both the activation and inactivation experiments (subsection “Optogenetic and pharmacological manipulation”). This is the maximum power provided in our current system. Because our experiments had a within-neuron control (i.e. the same cell without and with LED light illumination), variability in terminal power, expression level, etc. should not mask the overall effect.

The viral expression and fiber placement were always checked post hoc. Animals with improper locations of viral expression or fiber placement were excluded from data analysis (see the aforementioned subsection). These procedures help to minimalize potential confounding factors throughout our experiments.

5) The statistics used in the paper are insufficiently documented, and could be improved. In the Materials and methods section, it is stated that Shapiro-Wilk was used as a normality test prior to a paired t-test, if appropriate. However the results are never shown and the paired t-test seems to have been used for the majority of tests between on and off conditions throughout the paper. Please justify systematically the normality of the data or even better, use a non-parametric test (e.g. Signed test), which should just work as well.

In the Materials and methods, we have clarified that we used p value < 0.05 as a criterion for non-normality in the initial Shapiro-Wilk test. In the case of non-normality, we used the Wilcoxon signed-rank test as a non-parametric test (subsection “Statistics” and also specified in figure legends).

6) Things become a little bit more circuit unspecific later in the paper. For example, the visual looming experiment is only carried out with a mass pharmacological silencing of LP. The results are certainly suggestive, given the rest of the paper, but alternate circuits and mechanisms should be thoroughly discussed and acknowledged.

The reviewers have raised a good point. We have now discussed about possible alternative pathways, for example, in addition to the bottom-up SC-LP pathway, visual looming signals may reach LP through feedback input from visual and associative cortices and then modulate responses in A1 (subsection “LP mediates looming visual stimuli induced modulation of auditory responses in A1”). The experiment with pharmacological silencing of LP does not exclude the possibility of these alternative pathways.